# Aberrant Methylation of the Imprinted C19MC and MIR371-3 Clusters in Patients with Non-Small Cell Lung Cancer

**DOI:** 10.3390/cancers15051466

**Published:** 2023-02-25

**Authors:** Laura Boyero, José Francisco Noguera-Uclés, Alejandro Castillo-Peña, Ana Salinas, Amparo Sánchez-Gastaldo, Miriam Alonso, Johana Cristina Benedetti, Reyes Bernabé-Caro, Luis Paz-Ares, Sonia Molina-Pinelo

**Affiliations:** 1Institute of Biomedicine of Seville (IBiS), HUVR, CSIC, Universidad de Sevilla, 41013 Seville, Spain; 2Medical Oncology Department, Hospital Universitario Virgen del Rocío, 41013 Seville, Spain; 3H12O Lung Cancer Clinical Research Unit, Health Research Institute Hospital 12 de Octubre (imas12), 28029 Madrid, Spain; 4Spanish Center for Biomedical Research Network in Oncology (CIBERONC), 28029 Madrid, Spain; 5Spanish National Cancer Research Center (CNIO), 28029 Madrid, Spain; 6MD Anderson, 28033 Madrid, Spain

**Keywords:** DNA methylation, C19MC and MIR371-3 clusters, miRNA–target mRNA expression, prognosis, non-small cell lung cancer (NSCLC)

## Abstract

**Simple Summary:**

Aberrations in DNA methylation profiles may alter the expression of key miRNAs in non-small cell lung cancer. In this study, we focus on the analysis of the imprinted C19MC and MIR371-3 miRNA clusters due to their oncogenic role. We identified the DNA methylation status and discovered its deregulated target genes in this disease. Additionally, we found five downstream target genes that were correlated with worse overall survival in non-small cell lung cancer. We conclude that C19MC and MIR371-3 are key players in lung cancer because their polycistronic epigenetic regulation leads to differential tumor expression, affecting downstream targets with prognostic value.

**Abstract:**

Epigenetic mechanisms have emerged as an important contributor to tumor development through the modulation of gene expression. Our objective was to identify the methylation profile of the imprinted C19MC and MIR371-3 clusters in patients with non-small cell lung cancer (NSCLC) and to find their potential target genes, as well as to study their prognostic role. DNA methylation status was analyzed in a NSCLC patient cohort (*n* = 47) and compared with a control cohort including COPD patients and non-COPD subjects (*n* = 23) using the Illumina Infinium Human Methylation 450 BeadChip. Hypomethylation of miRNAs located on chromosome 19q13.42 was found to be specific for tumor tissue. We then identified the target mRNA–miRNA regulatory network for the components of the C19MC and MIR371-3 clusters using the miRTargetLink 2.0 Human tool. The correlations of miRNA-target mRNA expression from primary lung tumors were analyzed using the CancerMIRNome tool. From those negative correlations identified, we found that a lower expression of 5 of the target genes (*FOXF2*, *KLF13*, *MICA*, *TCEAL1* and *TGFBR2*) was significantly associated with poor overall survival. Taken together, this study demonstrates that the imprinted C19MC and MIR371-3 miRNA clusters undergo polycistronic epigenetic regulation leading to deregulation of important and common target genes with potential prognostic value in lung cancer.

## 1. Introduction

Lung cancer remains the most common cause of mortality worldwide, and tobacco exposure increases its risk of developing [1]. Furthermore, the incidence of lung cancer is significantly higher in patients diagnosed with chronic obstructive lung disease (COPD), reflecting the impact of smoking habits in both pathologies [2,3]. Approximately 85% of all lung cancers are non-small cell lung cancer (NSCLC). Histologically, NSCLC is classified as adenocarcinoma (ADC), squamous cell carcinoma and large cell carcinoma (SCC) [4]. Molecular analyses have led to advances in our understanding of NSCLC genetics and even in the identification of biomarkers that can predict its occurrence [5]. This includes the role of microRNAs (miRNAs) in the disease, involved in the complexity of gene expression regulation. Therefore, a single miRNA can exert its regulatory function on several target mRNAs, and a particular target can be regulated by multiple miRNAs [6].

There are numerous miRNAs involved in cancer-relevant processes, and many of them are clustered on the genome and act in coordinated regulatory networks. Furthermore, some evidence has even been provided suggesting that several miRNAs are able to identify patients with an increased risk of developing lung cancer, as well as COPD [7,8]. For a more extensive review of oncomiRs in lung cancer, see [9,10]. In this context, epigenetics appears to play an important role in the regulation of miRNA expression levels [11]. Aberrations in methylation profiles can promote silencing of tumor suppressor microRNAs or overexpression of oncogenic miRNAs (oncomiRs) [12]. For example, a significant upregulation of the miR-17-92 cluster has been reported in lung cancer [13]. In addition, some oncomiRs can be located in imprinted genomic regions. Such is the case of the imprinted delta-like homolog 1 gene and the type III iodothyronine deiodinase gene (DLK1-DIO3) cluster, which includes two large miRNA clusters between other coding and non-coding transcripts, and has been reported to contribute to tumorigenesis in the lung, leukemia, breast, and hepatoblastoma, among others [14,15,16,17,18]. In addition, alterations in other clusters located in imprinted regions are also attracting interest due to their downstream targets and their involvement in oncogenic and drug resistance mechanisms, such as the chromosome 19 microRNA (C19MC) and MIR371-3 clusters [19,20,21].

The C19MC and MIR371-3 clusters are located on chromosome 19q13.42. The first cluster includes forty-six miRNA genes and the second one contains four miRNAs (*miR-371*, *miR-372*, *miR-373* and *miR-373**). Both clusters are only expressed from their paternal allele, so they are functionally haploid and, furthermore, they are expressed mainly in embryonic tissue, particularly in the placenta [22,23]. However, aberrations that involve some miRNAs from both these clusters have been linked to tumoral processes, such as immunomodulation, angiogenesis, invasion, and cell reprograming [21,24,25,26,27,28,29]. In fact, some miRNAs through exosomes have been proposed as specific cell-to-cell communication mediators [21,27,30]. Regarding lung cancer, the regulation of imprinted C19MC and MIR371-3 has not been extensively and systematically reviewed in patients with NSCLC. For this reason, we have analyzed the methylation profile of the C19MC and MIR371-3 clusters in lung tumors compared to non-tumoral lung tissue in NSCLC patients. We have also assessed the methylation patterns of both clusters in COPD patients to study their association in a population at high risk of developing lung cancer. In addition, we were able to experimentally identify and validate deregulated targets, as well as their prognostic role in the disease.

## 2. Materials and Methods

### 2.1. Patients and Clinical Specimens

The present study was carried out on 70 subjects from the Virgen del Rocio University Hospital (Seville, Spain). Samples were divided into 2 cohorts according to the underlying pathology. The first cohort consisted of 47 NSCLC patients who had undergone surgical resection at an early clinical stage. During surgical resection, adjacent normal and tumor tissue samples were collected from all patients and immediately frozen at −80 °C until further use. The clinical characteristics of patients with NSCLC (*n* = 47) are summarized in Appendix A. The second cohort was used as control without lung cancer (*n* = 23). This control cohort consisted of COPD patients and non-COPD subjects (Appendix A) who had undergone bullectomy or bronchoscopic biopsy with a negative diagnosis of lung cancer. Both cohorts were used for the analysis of the methylation profile. The protocol of the study and the use of human samples were approved by the Ethics Committee of our hospital (1381-N-21). Written informed consent was obtained from all patients included in the study.

### 2.2. DNA Sample

Genomic DNA was extracted from 15 mg adjacent normal and tumor tissue samples using the QIAamp DNA mini kit (QIAGEN, Hilden, Germany). DNA was quantified using the QuantiFluor dsDNA system (Promega, Madison, WI, USA) according to the manufacturer’s instructions.

### 2.3. Illumina 450 K Methylation Assay

DNA methylation status at the CpG sites within the C19MC and MIR371-3 clusters was identified using the Illumina Infinium Human Methylation 450 BeadChip (Illumina Inc., San Diego, CA, USA). 500 ng of DNA were treated with sodium bisulfate using the EZ DNA Methylation™ Kit and cleaned with the ZR-96 DNA Clean-up Kit™ (Zymo Research, Irvine, CA, USA). Subsequently, the following steps were performed: amplification, hybridization and imaging. Intensity data was analyzed with Illumina’s GenomeStudio, from which, β-scores (i.e., the proportion of total fluorescence signal from the methylation-specific probe or color channel) were obtained. Infinium HD-based assays included sample-dependent and sample-independent controls for the highest quality data.

### 2.4. Methylome Data Processing

The methylome data was processed using the R/Bioconductor package RnBeads [31]. After a quality check, intensity normalization was performed by SWAN method [32] and converted to β values. The probes were tested for differential methylation with the limma linear model followed by empirical Bayes methods for the comparisons of interest [33]. Statistical significance was established using the Benjamini–Hochberg false discovery rate (FDR) with a value lower than 0.05. The DNA methylation status and CpG chromosomal location were displayed using the Circos software [34]. Furthermore, the methylation data was visualized by the Wash U Epigenome Browser [35].

### 2.5. Integrated Analysis of the Target mRNA–miRNA Regulatory Network

Strong validated miRNA–mRNA interactions were identified for the C19MC and MIR371-3 miRNA clusters with the miRTargetLink 2.0 Human tool. miRTargetLink collects information from various databases, including the sources miRBase (v.22.1) and miRTarBase (v.8) [36,37]. Gene expression analysis on tumor and normal lung tissue were analyzed from the Cancer Genome Atlas datasets (TCGA) using GEPIA 2.0 (http://gepia2.cancer-pku.cn/#index, accessed on 6 September 2021). GEPIA is an interactive web server that compiles data from TCGA and GTEx projects, using a standard processing pipeline. This tool allows for a customizable analysis of the collected data [38]. The molecular function and proposed biological process of the experimentally verified mRNAs were determined using the PANTHER program (http://pantherdb.org, accessed on 29 September 2021). The PANTHER classification system contains a comprehensive, annotated “library” of phylogenetic trees of gene families designed to classify proteins (and their genes) to facilitate high-throughput analysis [39]. Besides, Kyoto Encyclopedia of Genes and Genomes (KEGG) pathway database was used to know the molecular interaction network (https://www.genome.jp/kegg/pathway.html, accessed on 3 October 2021) [40].

### 2.6. Correlation Analysis in Lung Primary Tumors from TCGA Datasets

Transcriptome profiling data from TCGA datasets were downloaded using the CancerMIRNome tool [41]. MicroRNA and mRNA expression data from primary tumors were retrieved from the LUAD (Lung Adenocarcinoma) and LUSC (Lung Squamous Carcinoma) datasets included in TCGA. Only samples labeled as “tumor” and expression level (Log_2_ Counts per Million (CPM)) > −3.322 were used for the analyses. To test the association between paired miRNA–mRNA profiles, the Pearson correlation coefficients and *p*-values were computed. *p*-values lower than 0.05 were considered statistically significant.

### 2.7. Survival Analysis to Assess the Prognostic Value of Validated Target Genes of the C19MC and MIR371-3 miRNA Clusters in NSCLC Patients

To analyze the prognosis associated with the target genes of the C19MC and MIR371-3 miRNA clusters, the Kaplan–Meier survival plots to overall survival time were obtained using the Kaplan–Meier (KM) plotter website [42], where unprocessed. CEL files from the Gene Expression Omnibus (GEO), the European Genome-phenome Archive (EGA) and the Cancer Genome Atlas (TCGA) repositories were normalized in the R environment. The datasets included in the Kaplan–Meier plotter website are GSE3141, GSE4573, GSE8894, GSE14814, GSE19188, GSE29013, GSE31210, GSE37745, EGA and TCGA (*n* = 1715 patients). The best-performing threshold from computed lower and upper quartiles was used as cut-off point for the definition of high and low expression of the analyzed genes. Overall survival (OS) was determined from the date of diagnosis to the date of death. *p*-values lower than 0.05 were considered statistically significant.

## 3. Results

### 3.1. DNA Methylation Pattern of the C19MC and MIR371-3 miRNA Clusters in Lung Cancer

To evaluate the potential role of the C19MC and MIR371-3 miRNA clusters in lung cancer, we analyzed the methylation status of both of them in human lung tissues from a NSCLC patient cohort (*n* = 47) and a control cohort (*n* = 23) of the Virgen del Rocio University Hospital (Seville, Spain). The methylation profile of these clustered miRNAs, which are located on chromosome 19q13.42, were evaluated in human tumor samples compared to paired non-tumoral tissue by using the Illumina Infinium Human Methylation 450 BeadChip. The methylation levels in lung cancer versus paired non-tumoral tissues are represented in Figure 1A and Appendix A. Patients with lung cancer at our hospital showed DNA hypomethylation at 50 miRNAs included in the C19MC and MIR371-3 clusters after standardisation with non-tumoral control samples. Statistically significant differences (adjusted *p*-value < 0.05) were detected in the large C19MC cluster (46 miRNAs) and the closely distal MIR371–3 cluster (*miR-371a*, *miR-371b*, *miR-372* and *miR-373*). Among all miRNAs, *miR-520b*, *miR-520c*, *miR-520f*, *miR-526a1*, *miR-1283-1* and *miR-1283-2* showed greater changes in the DNA hypomethylation pattern in NSCLC patients (Figure 1B,C and Table 1).

We next analyzed the DNA methylation pattern of both clusters in patients at high risk of developing lung cancer, such as patients with COPD. In these patients, we found that the C19MC cluster methylation profile showed no statistical differences compared to the control group (Figure 1B and Appendix A). We even found that *miRNA-520e*, *miR-524* and *miR-516b2* were hypermethylated in COPD patients versus the non-tumoral control group (Figure 1B). However, these differences did not reach statistical significance. In the case of the MIR371-3 cluster, changes in the DNA methylation levels were negligible between both patients groups (Figure 1C and Appendix A).

### 3.2. Transcriptional Mapping of the C19MC and MIR371-3 miRNA Clusters

To analyze genomic features associated with different mechanisms of the transcriptional regulation of the C19MC and MIR371-3 miRNA clusters, we used the Wash U Epigenome Browser [33] to display the epigenomic mapping of both clusters (19q13.42) on the reference human genome (hg19; chr19: 54,030,000–54,430,000 genomic coordinates) (Figure 2A). Thus, it is shown which transcriptional mechanisms (direct regulators and structural determinants) act at each locus throughout both clusters (Figure 2B). We observed that enhancers are distributed outside the regions where the C19MC and MIR371-3 miRNA clusters are located (yellow histograms). In these same regions, we found low activity of polycomb protein-mediated epigenetic regulator (grey histograms). In addition, we identified two heterochromatin-rich domains located in the C19MC cluster region (purple histograms). One of these highly condensed regions is observed at 5′ of the C19MC miRNA cluster. Interestingly, the 5′ and 3′ flanking regions of both clusters showed transcription initiation activity, marked by small active transcriptional start sites (TSS) shown by red histograms. No other region of the C19MC and MIR371-3 miRNA clusters showed active TSS. Behind these active TSS, we found regions with strong transcription activity (green histograms) (Figure 2B). We identified a CpG island in the 5′ region at the beginning of the C19MC miRNA cluster (~54,150,000 bp; CpG count: 86; and citosine base count plus guanine base count: 762) and another in the MIR371-3 miRNA cluster (~54,270,000 bp; CpG count: 23; and citosine count plus guanine count: 157) (green lines) (Figure 2C). However, the CpG sites were distributed throughout both clusters (white lines) (Figure 2D). The CG-content was similar from 54,030,000 to 54,430,000 bp on chromosome 19 (Figure 2E).

### 3.3. Methylation Profile of the C19MC and MIR371-3 miRNA Clusters by Histological Subtypes

Due to lung cancer-specific hypomethylation, in order to evaluate the potential role of the C19MC and MIR371-3 miRNA clusters as biomarkers in different histological subtypes of lung cancer, we analyzed the methylation pattern in SCC versus adenocarcinoma, pre-normalizing each patient with the methylation status of the matched non-tumoral tissue. The DNA-methylation levels of the C19MC and MIR371-3 clusters were consistently hypomethylated in both histological subtypes in comparison with non-tumoral tissue (Appendix A). Furthermore, these differences were statistically significant for both clusters (C19MC, *p* < 0.001; MIR371-2, *p* = 0.030) (Figure 3).

### 3.4. Experimentally Validated miRNA–Target Interactions for the C19MC and MIR371-2 miRNA Clusters

We have graphically represented a network with experimentally validated miRNA–mRNA interactions for each component of the C19MC and MIR371-3 clusters in order to study their functional relevance (Figure 4). According to the data included in miRTargetLink software [36] and considering only strong evidence targets, a total of 115 genes were found targeted by at least one of the aforementioned miRNAs. In the network of the C19MC miRNA cluster, the nodes with several connections were those corresponding to *miRNA-512-5p*, *miR-518a-5p*, *miR-519b-3p*, *miR-519a-3p*, *miR-519d-3p*, *miR-520a-3p*, *miR-520c-3p*, *miR-520g-3p*, *miR-520h*, *miR-524-5p* and *miR-525-3p* (Figure 4A). Among the MIR371-3 cluster compounds, *miR-371a-3p*, *miR-372-3p* and *miR-373-3p* showed a high number of connections. The common target genes possessed a high interaction grade with different miRNAs simultaneously (Figure 4B and Table 2). Of these targets, we identified six genes (*CD44*, *CDKN1A*, *MTOR*, *SIRT1*, *TGFBR2* and *VEGFA*) that were targeted by miRNAs from both clusters.

### 3.5. Aberrant Expression of Validated Target Genes for Both miRNA Clusters in Lung Cancer Patients

We explored the transcriptional levels of the previously identified validated target genes in cancer and normal lung tissues in the TCGA/GTEx data available in GEPIA2 [38]. Of the 115 target genes validated by miRTargetLink, 31 were significantly underexpressed in at least one of the histological subtypes of NSCLC compared to non-tumor tissue (Table 3 and Figure 5). Seventeen of them showed significant differences in both histological subtypes of lung cancer (adenocarcinoma and SCC). It is worth highlighting those genes that we found targeted by several miRNAs from both clusters, such as the tumor suppressor genes (TSG) *CDKN1A* (regulated by *miR-372-3p*, *miR-512-5p*, *miR-515-3p*, *miR-519a-3p*, *miR-519b-3p*, *miR-519e-3p* and *miR-520a-3p*) and *TGFBR2* (targeted by *miRNA-372-3p*, *miRN-373-3p* and *miR-520a-3p*).

As displayed in Figure 6, we classified the validated targets by gene ontology (GO) molecular function and biological processes using the PANTHER software [39]. The main GO molecular functions were binding (GO:0005488) (39.0%), catalytic activity (GO:0003824) (26.8%) and transcription regulator activity (GO:0001071) (14.6%). The primary binding types were protein binding (GO:0005515) and organic cyclic compound binding (GO:0097159). In the case of catalytic activity, we primarily found hydrolase (GO:0016787), transferase (GO:0016740) and catalytic (GO:0140096) activities. Validated targets displayed three main biological processes: response to stimulus (GO:0050896) (15.0%), biological regulation (GO:0065007) (17.7%) and cellular process (GO:0009987) (19.5%). In the latter, we found represented cell communication (GO:0030234), signal transduction (GO:0007165) and cellular metabolic process (GO:0044237). Finally, we found that 31% of validated targets with aberrant expression in lung cancer were classified in cancer by the KEGG pathways (adjusted *p* = 7.0 × 10^−3^).

### 3.6. Correlation of the Validated Target mRNA–miRNA Expression in Lung Primary Tumors

Since we had compelling data on the functional relationship between miRNA–target for the C19MC and MIR371-3 miRNA clusters, as well as differential expression of the validated target genes in lung cancer, we tested whether miRNA–target correlations extended to primary tumors. We studied these correlations using external expression data obtained from the TCGA datasets through the CancerMIRNome tool [41]. We found 11 significant negative correlations for lung adenocarcinoma (Figure 7A), all of them for miRNAs included only in the C19MC cluster. On the other hand, 8 were found for SCC (Figure 7B), this time including miRNAs from both clusters. It should be noted that *CDKN1A* was the target gene with the highest number of significant negative correlations (*p* < 0.05) in both adenocarcinoma (*miRNA-512-5p*, *miRNA-512-3p*, *miRNA-520a-3p* and *miRNA-520h*) and SCC (*miRNA-515-3p*, *miRNA-519b-3p*, *miRNA-520a-3p* and *mir-372-3p*). Furthermore, *DAPK2* was also negatively correlated with *miRNA-520g-3p* and *miRNA-520h* in both histological subtypes of lung cancer (*p* < 0.01). Other significant correlations for adenocarcinoma were *PTK2B-miRNA-517c-3p* (*p* < 0.001), *FOXF2-miRNA-519a-3p* (*p* = 0.001), *TGFBR2-miRNA-520a-3p* (*p* = 0.009), *MICA-miRNA-520c-3p* (*p* = 0.007) and *TECEAL1-miRNA-520g-3p* (*p* = 0.037); while for SCC, they were *JAG1-miRNA-524-5p* (*p* < 0.001) and *KLF13-miRNA-372-3p* (*p* = 0.003).

### 3.7. Prognostic Role of the Target Gene Network of the C19MC and MIR371-3 Clusters in Lung Cancer

To evaluate whether the genes targeted by the C19MC and MIR371-3 clusters with significant negative correlations were associated with clinical outcomes in patients with lung cancer, we analyzed their gene expression levels according to OS data using the KM Plotter website [42]. We found that five target genes were also significantly associated with worsening OS (Figure 8). These genes were *FOXF2* (HR = 0.66, 95% CI = 0.58–0.75, *p* < 0.001), *KLF13* (HR = 0.61, 95% CI = 0.52–0.72, *p* < 0.001), *MICA* (HR = 0.75, 95% CI = 0.66–0.86, *p* < 0.001), *TCEAL1* (HR = 0.72, 95% CI = 0.63–0.82, *p* < 0.001) and *TGFBR2* (HR = 0.66, 95% CI = 0.58–0.75, *p* < 0.001). The expression levels of the rest of the genes did not show significant differences regarding the OS in patients with lung cancer.

## 4. Discussion

In the present study, we identified the methylation pattern of the imprinted C19MC and MIR371-3 clusters in patients with NSCLC. Specifically, we identified that all compounds of both clusters are hypomethylated in tumor lung tissue compared to paired normal lung tissue. Importantly, this imprinted cluster-specific methylation signature is restricted to lung cancer, as it is absent in patients with COPD, who have an increased risk of developing lung cancer. Furthermore, these epigenetic changes observed in patients with NSCLC are negatively associated with the expression of relevant target genes in the disease; even five of them were significantly associated with the prognosis of this disease.

This methylation profile is consistent with the epigenetic features found at the 19q13.42 locus, in which, the C19MC and MIR371-3 miRNA clusters are located. Epigenetic modifications are characterized by not altering the nucleotide sequence, but by regulating genetic structure and expression reversibly. Early events in tumorigenesis and its progression are related to DNA methylation, histone modifications, nucleosome remodeling and miRNA, which are key epigenetic players [43]. In the particular case of C19MC and MIR371-3 miRNA clusters, the mechanisms involved in regulation were the presence of enhancers, CpG islands, active TSS, heterochromatin-rich domains, as well as low activity of polycomb complexes. In addition, these differences were evident for both histological subtypes of NSCLC. Thus, we found that although all subtypes are hypomethylated, this is more accentuated in SCC than in lung adenocarcinoma. At the same locus, the Protein Tyrosine Phosphatase Receptor type H (*PTPRH*) has also been confirmed to be hypomethylated and this is correlated with increased gene expression and leads to a poor prognosis in NSCLC [44,45]. Therefore, the epigenetic features of this region have a significant effect on the disease.

Members of the C19MC and MIR371-3 clusters are expressed almost exclusively in human embryonic stem cells (hESC) and rapidly down-regulated during the differentiation process [46,47,48]. However, they are over-expressed in cancer [20,48], suggesting a tumorigenic role and a possible maintenance function of tumor-associated progenitor cells for these clusters when reactivated [49]. Higher expression of members of the C19MC and MIR371-3 clusters has also been reported in thyroid adenomas [50] and parathyroid carcinomas [48], germ cell tumors [29,51,52], retinoblastoma [53], breast cancer [54], gastric adenocarcinoma [55] and esophageal cancer [56,57], among others. And this increase affects tumor growth, differentiation, progression and aggressiveness and, ultimately, patient survival. This cluster overexpression phenomenon also occurs in other sets of miRNAs in NSCLC, such as the miR-23a/27a/24-2 cluster, which has predictive value in early stages and stimulates postoperative progression by inducing tumor suppressor gene silencing [58]. Activation of all miRNAs from the DLK1-DIO3 locus has also been described for human lung adenocarcinoma samples, which is associated with cell stemness and its targets are involved in embryogenesis [59]. Moreover, it is hypomethylated in current and former smokers with NSCLC, suggesting a relevant role in the pathogenesis of lung cancer [15]. The 14q32 miRNA cluster is another example of up-regulation due to DNA hypomethylation in metastatic lung adenocarcinoma patients. Overexpression of this cluster induces cell migration and invasion and has prognostic value [60]. Furthermore, overexpression of the miR-17-92 cluster, which is a highly conserved oncogene cluster, has been frequently reported in lung cancer, especially in the small cell lung cancer subtype, promoting cell growth [61,62]. Therefore, the gene regulation mechanism mediated by miRNAs is very interesting due to its ability to associate with mRNAs of multiple targets.

On the other hand, we identified a total of 115 strongly validated targets for the C19MC or MIR371-3 clusters, and 31 of them presented a significant lower transcriptional level in NSCLC tissue compared to normal. In other words, we identified only those miRNA–mRNA interactions that had been experimentally validated previously. Of these, only six genes were targeted by members of both clusters: *CD44*, *CDKN1A*, *MTOR*, *SIRT1*, *TGFBR2* and *VEGFA*. In addition, these miRNA–target interactions were verified with TCGA data for primary NSCLC tumors, corroborating significant miRNA–target negative correlations in both clusters and histological subtypes of the disease. Interestingly, redundancy is observed in the miRNAs belonging to C19MC and MIR371-3, since many of them share targets, as it can be inferred from our results. An explanation for this is the high degree of homology that has been described for some C19MC miRNAs through a seed region (5′-AAGUGC-3’), which can be found in several members of the cluster at different positions [20,46]. This suggests that, in addition to a polycistronic regulation, the targets of these miRNAs are common or share functions. Bioinformatic predictions for this seed region relate these miRNAs to cellular proliferation and apoptosis [20,63]. Something similar occurs with the MIR371-3 cluster, which has identical seed sequences that are similar to its murine miR290-295 homolog [64].

In this study, we even found that this redundancy occurs between the two clusters C19MC and MIR371-3. The *CDKN1A* and *DAPK2* genes are notable for being negatively correlated with and targeted by multiple miRNAs from these clusters in the two most common subtypes of NSCLC (lung adenocarcinoma and SCC). *CDKN1A*, also known as *p21*, plays a critical role in the cellular response to DNA damage, and its overexpression results in p53-mediated cell cycle arrest [65]. It has been reported that *CDKN1A/p21* can be blocked in NSCLC by oncomiRs, such as *miR-212/132* [66] or *miR-93*. The latter can act directly or indirectly, thereby inhibiting liver kinase B1 (*LKB1*) [67]. This promotes proliferation and metastases through the phosphatidylinositol 3-kinase (*PI3K*)/protein kinase B (*AKT*) pathway in NSCLC. Other authors have also confirmed the relationship between *CDKN1A/p21* and members of the MIR371-3 cluster in hESCs [68], specifically *miR-372*. On the other hand, *DAPK2* is a serine/threonine kinase that promotes cell apoptosis and autophagy [69] by activating the oncogenic nuclear factor kappa-light-chain-enhancer of activated B cells (*NF-κB*) signaling pathway [70], which sensitizes resistant cells to tumor necrosis factor (*TNF*)-related apoptosis-inducing ligand (*TRAIL*)-mediated death [71]. *DAPK2* expression has been reported to be significantly associated with the poor prognosis in NSCLC [70,72]. As in this study, *DAPK2* is also downregulated by *miR-520h* in breast cancer [73] and miR-520g in epithelial ovarian cancer [74] and it contributes to chemoresistance.

Finally, we found that five of the genes targeted by members of the C19MC and MIR371-3 clusters correlated with worse OS in lung cancer. These genes were *FOXF2*, *KLF13*, *MICA*, *TCEAL1* and *TGFBR2*. They are known transcription regulators and oncogenes, some involved in immune evasion and resistance to immune checkpoint inhibitors [75], which places them in the focus of current cancer research. For example, according to our results, decreased *FOXF2* expression has been reported as an independent predictor of poor prognosis for patients with early-stage NSCLC [76]. *FOXF2* deregulation ought to an aberrant DNA methylation status has also recently been identified for gastric cancer [77]. In the case of *MICA*, Okita et al. reported that *PD-L1*^low^/*MICA*/*B*^high^ is associated with a better clinical outcome in patients with stage I-IIIA NSCLC. On the other hand, both oncogenic and tumor suppressor roles have been attributed to the transforming growth factor beta (*TGF-β*) pathway, depending on both the type of tumor and its stage [78]. In this matter, *TGFBR2* [79] has recently been proposed as a tumor suppressor with prognostic value in early-stage NSCLC; however, there is no prior evidence of an association between *KLF13* and lung cancer prognosis, so further research is required.

Our study has the limitation that the mechanism underlying the alterated methylation status of the C19MC- and MIR371-3-imprinted clusters was not evaluated. Cancer-related gene hypomethylation is common in solid tumors, which could contribute to increased expression of oncogenes. In this case, miRNAs may act as oncomiRs by inhibiting the expression of tumor suppressor genes. DNA methylation is a dynamic process regulated by the action of DNA demethylases and DNA methyltransferases (DNMT) [80]. Alterations in the expression or activity of these enzymes can lead to changes in DNA methylation patterns that can trigger alterations in key genes involved in cancer development. For example, Zhang et al. have reported that *DNMT1*, *DNMT3A* and *DNMT3B* show frequency alterations in approximately 3% to 5% of lung cancer patients from the cBioPortal datasets [81]. However, further research is needed to validate the mechanisms in detail, e.g., in organoid in vitro and/or in preclinical in vivo models, to fully understand the mechanisms upstream involved in the aberrant methylation of both clusters and to assess the potential therapeutic applications of targeting them. In addition, we identify relevant miRNA–mRNA interactions in NSCLC patients. In other words, the regulation of the expression of these genes can be partly explained by the activity of these miRNAs; nevertheless, it should be noted that additional mechanisms may also be involved in the regulation of these genes, such as deletion, amplification, mutation, fusion and multiple alterations, and even other miRNAs not included in these clusters, as their regulation can be mediated by the action of many miRNAs. Despite these limitations, this study provides important insights into the consequences of these epigenetic alterations in NSCLC patients and highlights potential targets for future research and therapy. On the other hand, another limitation would be the sample size of patients with COPD included in the study. We have analyzed the DNA methylation pattern of the C19MC and MIR371-3 clusters in a group of COPD patients because they have a three- to six-fold increased risk of developing lung cancer, even if they quit smoking [2]. In our study, we found no statistical differences between the control group without lung cancer and COPD. However, sample size is an important consideration because it may affect the accuracy and reliability of the results. A larger size should be considered to conclude the effect of methylation status of both clusters in this disease.

## 5. Conclusions

In conclusion, this study demonstrates that the imprinted C19MC and MIR371-3 clusters undergo polycistronic epigenetic regulation that leads to differential tumor expression in NSCLC patients. These differences, in turn, deregulate the expression of important and common target genes, many of them with a clear oncogenic and regulatory role in this disease, highlighting five genes (*FOXF2*, *KLF13*, *MICA*, *TCEAL1*, *TGFBR2*) that have also a potential prognostic value in NSCLC patients. All these characteristics make the different components of both clusters an interesting target in oncology that needs to be further investigated; it would even be interesting to evaluate the use of methylation agents as an alternative approach to lung cancer therapy.

## Figures and Tables

**Figure 1 cancers-15-01466-f001:**
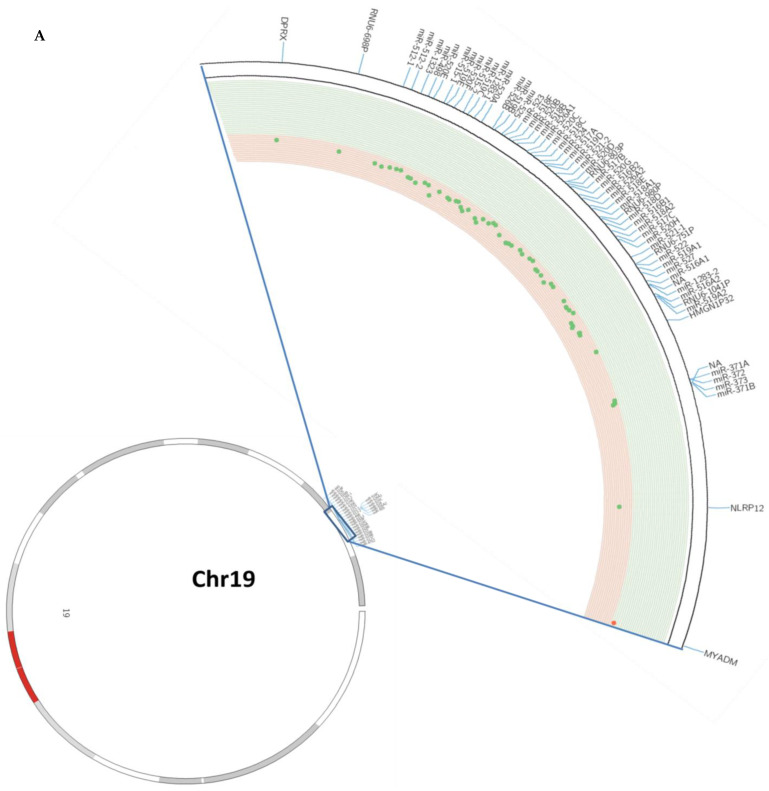
Methylation profiles of the C19MC and MIR371-3 miRNA clusters in lung cancer. (**A**) Circos plot showing the methylation levels on chromosome 19. From inside to outside: methylation levels, ideogram and gene labels. Hypermethylation (red dots and green background) and hypomethylation events (green dots and red background) in patients with lung cancer versus paired non-tumoral samples. NA: Not available. (**B**) Observed methylation changes (log_2_ ratio) in the C19MC cluster. Relative levels of methylation in patients with lung cancer relative to the control group are represented in blue bars, whereas methylation levels of COPD patients without lung cancer compared to the non-tumoral control group are represented by red bars. A grey background represents statistically significant differences (adjusted *p*-value < 0.05) of methylation levels relative to the control group. (**C**) Observed methylation changes (log_2_ ratio) in the MIR371-3 cluster. Relative levels of methylation in patients with lung cancer relative to the control group are represented in blue bars, whereas methylation levels of COPD patients without lung cancer compared to the non-tumoral control group are represented by red bars. A grey background represents statistically significant differences (adjusted *p*-value < 0.05) of methylation levels in comparison to the non-tumoral control group.

**Figure 2 cancers-15-01466-f002:**
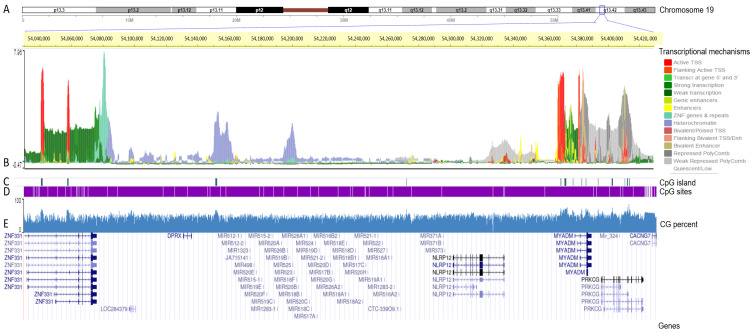
Transcriptional mapping of the C19MC and MIR371-3 clusters on the reference human genome hg19. (**A**) Chromosome 19 ideogram. The C19MC and MIR371-3 clusters are located on 19q13.42. The exact position of the cluster in the region is marked with a blue square. (**B**) Chromosome position. Base pairs of the C19MC and MIR371-3 clusters on chromosome 19 (highlighted in yellow). Transcriptional mechanisms underlying the expression of both clusters, such as transcription start sites (TSS), enhancer regions (Enh), zinc finger (ZNF), packed form of DNA and polycomb group proteins. (**C**) CpG islands in the C19MC and MIR371-3 clusters (green lines). (**D**) Genomic distribution of CpG sites in the C19MC and MIR371-3 clusters (white lines). (**E**) CG percentage in the C19MC and MIR371-3 clusters (blue). Finally, the reference sequences of the members of the C19MC and MIR371-3 cluster are represented at the bottom.

**Figure 3 cancers-15-01466-f003:**
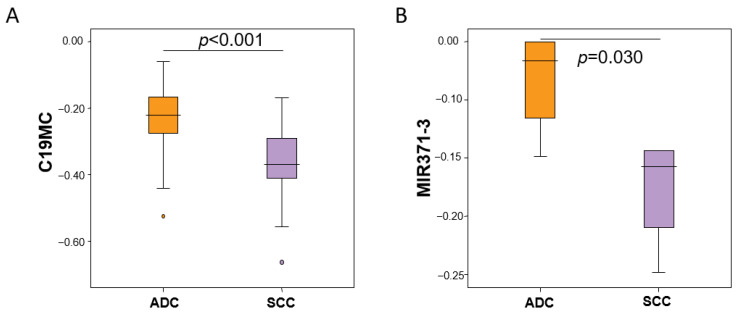
Methylation levels of the C19MC and MIR371-3 clusters components according to the histological subtype of lung cancer. (**A**) C19MC methylation levels in patients with squamous cell carcinoma (SCC) relative to the control group is represented in purple bars, whereas methylation levels in patients with adenocarcinoma (ADC) compared to the control group are represented in orange bars. (**B**) Relative MIR371-3 methylation levels in patients with SCC (purple) and ADC (orange) relative to the control group. Relative methylation changes in β values (log_2_) are represented on the *y*-axis.

**Figure 4 cancers-15-01466-f004:**
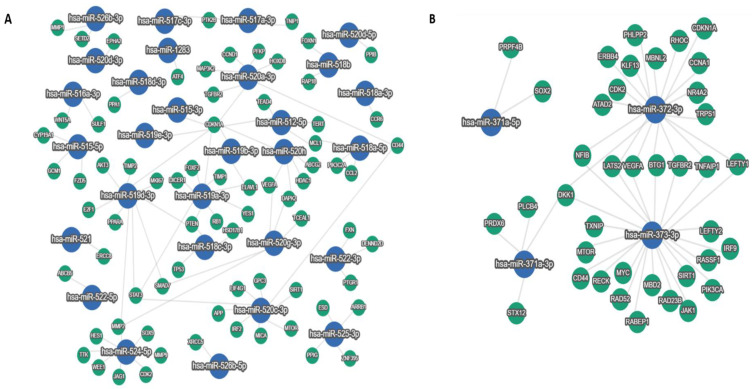
miRNAs–target genes network. Strong interactions are displayed for experimentally validated target genes and the components of the C19MC (**A**) and MIR371-3. (**B**) clusters generated by miRTargetLink 2.0. Network layout: Random. Nodes are represented as circles: miRNAs (blue) and target genes (green).

**Figure 5 cancers-15-01466-f005:**
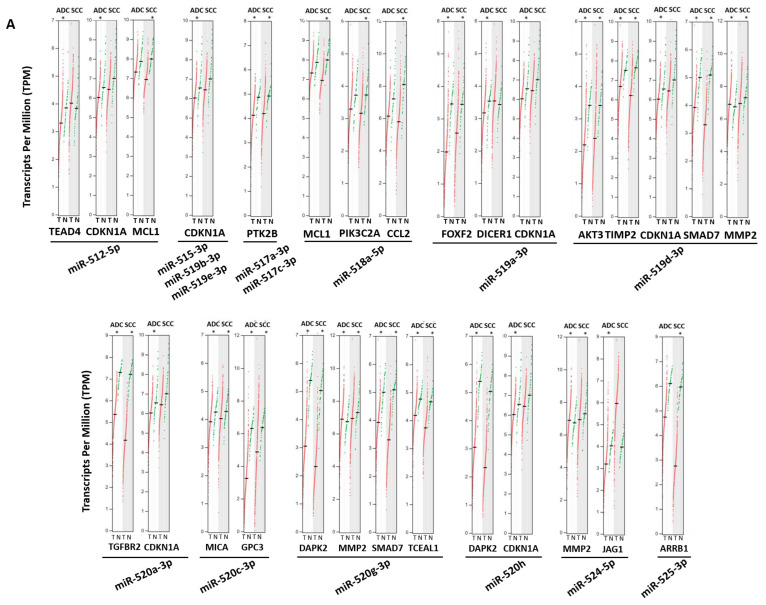
Transcripts per million (TPM) in normal (green) and tumor (red) lung tissues of those genes identified as strongly validated miRNA targets. (**A**) C19MC cluster and (**B**) MIR371-3 cluster. T: Tumor Tissue; N: Normal Tissue; ADC: Adenocarcinoma Lung; SCC: Squamous Cell Carcinoma. * ANOVA test with *p*-value < 0.01.

**Figure 6 cancers-15-01466-f006:**
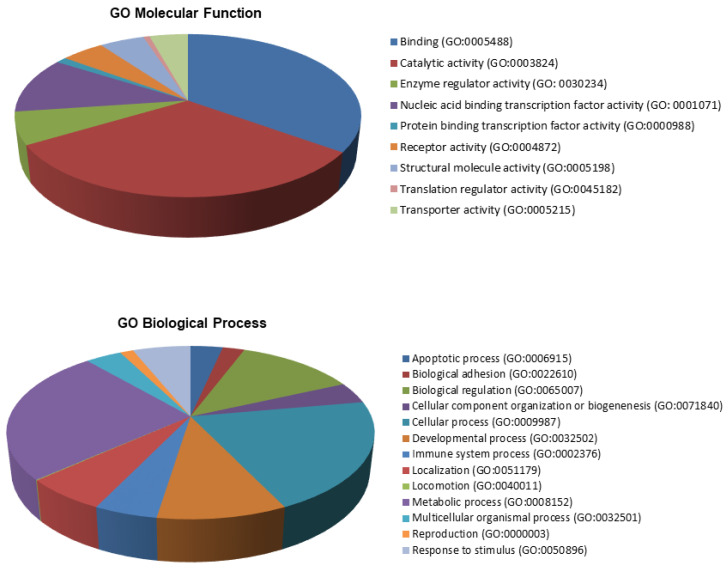
Classification of validated targets by gene ontology (GO) molecular function and biological processes. Summary of molecular functions and biological processes for validated target genes of miRNAs included in both clusters.

**Figure 7 cancers-15-01466-f007:**
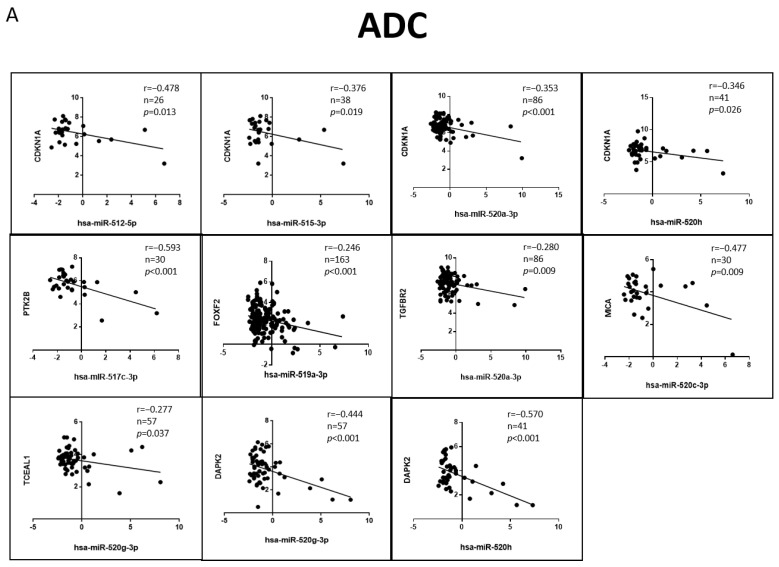
miRNA–target correlations in lung primary tumors. Significant negative correlations between miRNA–target for the C19MC and MIR371-3 miRNA clusters in (**A**) ADC and (**B**) SCC lung primary tumors. r: Pearson’s correlation coefficient; n: sample size, ADC: Lung Adenocarcinoma; SCC: Lung Squamous. *p*-values lower than 0.05 were considered statistically significant.

**Figure 8 cancers-15-01466-f008:**
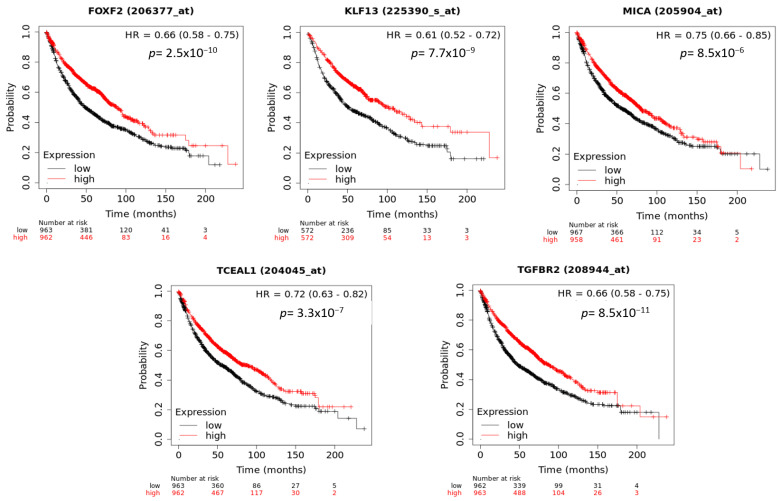
Clinical outcomes significantly associated with target gene expression in lung cancer. The Kaplan–Meier survival plots were obtained using the Kaplan–Meier plotter website. HR: hazard ratio.

**Table 1 cancers-15-01466-t001:** Statistically differences of the methylation levels of C19MC and MIR371-3 clusters between human lung tumor samples and normal lung.

	Genes	Relative Methylation Changes (log_2_)	Adjusted *p*-Value
C19MC	*miR-512-1*	−0.307	6.3946 × 10^−14^
	*miR-512-2*	−0.341	7.2597 × 10^−9^
	*miR-1323*	−0.245	1.0347 × 10^−6^
	*miR-498*	−0.241	2.7704 × 10^−5^
	*miR-520e*	−0.214	5.7827 × 10^−5^
	*miR-515-1*	−0.346	2.0661 × 10^−12^
	*miR-519e*	−0.340	3.1486 × 10^−11^
	*miR-520f*	−0.402	4.644 × 10^−13^
	*miR-515-2*	−0.258	2.686 × 10^−9^
	*miR-519c*	−0.264	2.9066 × 10^−10^
	*miR-1283-1*	−0.601	4.8535 × 10^−17^
	*miR-520a*	−0.279	4.1063 × 10^−11^
	*miR-526b*	−0.329	5.2657 × 10^−14^
	*miR-519b*	−0.348	1.5969 × 10^−15^
	*miR-525*	−0.360	1.2282 × 10^−10^
	*miR-523*	−0.267	1.1234 × 10^−11^
	*miR-518f*	−0.368	7.9986 × 10^−15^
	*miR-520b*	−0.441	4.5322 × 10^−16^
	*miR-518b*	−0.290	1.7394 × 10^−9^
	*miR-526a1*	−0.474	3.4512 × 10^−14^
	*miR-520c*	−0.590	4.7234 × 10^−18^
	*miR-518c*	−0.271	6.5541 × 10^−9^
	*miR-524*	−0.261	1.5518 × 10^−12^
	*miR-517a*	−0.129	1.6843 × 10^−5^
	*miR-519d*	−0.140	3.7082 × 10^−5^
	*miR-521-2*	−0.343	1.1838 × 10^−14^
	*miR-520d*	−0.378	2.9354 × 10^−13^
	*miR-517b*	−0.327	2.6624 × 10^−12^
	*miR-520g*	−0.293	9.6092 × 10^−11^
	*miR-516b2*	−0.215	1.1738 × 10^−8^
	*miR-526a2*	−0.303	9.5413 × 10^−10^
	*miR-518e*	−0.168	6.8329 × 10^−7^
	*miR-518a1*	−0.139	7.0465 × 10^−5^
	*miR-518d*	−0.412	2.144 × 10^−12^
	*miR-516b1*	−0.269	9.8287 × 10^−10^
	*miR-518a2*	−0.364	2.9253 × 10^−14^
	*miR-517c*	−0.199	1.476 × 10^−10^
	*miR-520h*	−0.211	2.5861 × 10^−10^
	*miR-521-1*	−0.183	8.9968 × 10^−5^
	*miR-522*	−0.235	5.605 × 10^−10^
	*miR-519a1*	−0.217	9.8489 × 10^−11^
	*miR-527*	−0.130	7.4475 × 10^−9^
	*miR-516a1*	−0.400	2.5393 × 10^−9^
	*miR-1283-2*	−0.453	2.9539 × 10^−13^
	*miR-516a2*	−0.340	3.3131 × 10^−16^
	*miR-519a2*	−0.344	1.2518 × 10^−14^
MIR371-3	*miR-371b*	−0.209	1.4784 × 10^−8^
	*miR-371a*	−0.106	7.6971 × 10^−7^
	*miR-372*	−0.106	7.6971 × 10^−7^
	*miR-373*	−0.137	1.183 × 10^−7^

**Table 2 cancers-15-01466-t002:** Common miRNA–target interactions for both C19MC and MIR371-2 miRNA clusters.

Gene	Gene Description	miRNA
*BTG1*	B-Cell Translocation Gene 1	*miRNA-372-3p*, *miRNA-373-3p*
*CD44*	Cluster of Differentiation 44	*miRNA-520a-3p*, *miRNA-520c-3p*, *miRNA-373-3p*
*CDK2*	Cyclin Dependent Kinase 2	*miRNA-524-5p*, *miRNA-372-3p*
*CDKN1A*	Cyclin Dependent Kinase Inhibitor 1A	*miRNA-512-5p*, *miRNA-515-3p*, *miRNA-519a-3p*, *miRNA-519b-3p*, *miRNA-519d-3p*, *miRNA-519e-3p*, *miRNA-520a-3p*, *miRNA-520h, miRNA-373-3p*
*DAPK2*	Death Associated Protein Kinase 2	*miRNA-520h, miRNA-520g-3p*
*DKK1*	Dickkopf WNT Signaling Pathway Inhibitor 1	*miRNA-371a-3p*, *miRNA-372-3p*, *miRNA-373-3p*
*ELAVL1*	ELAV (Embryonic Lethal, Abnormal Vision, Drosophila)-Like RNA Binding Protein 1	*miRNA-519a-3p*, *miRNA-519b-3p*
*LATS2*	Large Tumor Suppressor Kinase 2	*miRNA-372-3p*, *miRNA-373-3p*
*LEFTY1*	Left-Right Determination Factor 1	*miRNA-372-3p*, *miRNA-373-3p*
*MCL1*	Myeloid Cell Leukemia Sequence 1	*miRNA-512-5p*, *miRNA-518a-5p*
*MMP2*	Matrix Metallopeptidase 2	*miRNA-519d-3p*, *miRNA-520g-3p*, *miRNA-524-5p*
*MTOR*	Mechanistic Target of Rapamycin Kinase	*miRNA-520c-3p*, *miRNA-373-3p*
*NFIB*	Nuclear Factor I B	*miRNA-372-3p*, *miRNA-373-3p*
*PTEN*	Phosphatase and Tensin Homolog	*miRNA-518c-3p*, *miRNA-519a-3p*, *miRNA-519d-3p*
*PTK2B*	Protein Tyrosine Kinase 2 Beta	*miRNA-517a-3p*, *miRNA-517c-3p*
*SIRT1*	Sirtuin 1	*miRNA-520c-3p*, *miRNA-373-3p*
*SAMD7*	Sterile Alpha Motif Domain Containing 7	*miRNA-519d-3p*, *miRNA-520g-3p*
*STAT3*	Signal Transducer and Activator of Transcription 3	*miRNA-519a-3p*, *miRNA-519g-3p*, *miRNA-520c-3p*
*TGFBR2*	Transforming Growth Factor Beta Receptor 2	*miRNA-520a-3p*, *miRNA-372-3p*, *miRNA-373-3p*
*TNFAIP1*	Tumor Necrosis Factor Alpha-Induced Protein 1	*miRNA-372-3p*, *miRNA-373-3p*
*VEGFA*	Vascular Endothelial Growth Factor A	*miRNA-520g-3p*, *miRNA-520h, miRNA-372-3p*, *miRNA-373-3p*

**Table 3 cancers-15-01466-t003:** Target genes with significant expression differences in NSCLC tissue compared to normal lung tissue.

Symbol	Description
*ARRB1*	Arrestin Beta 1
*AKT3*	AKT Serine/Threonine Kinase 3
*CCL2*	C-C Motif Chemokine Ligand 2
*CDKN1A*	Cyclin Dependent Kinase Inhibitor 1A
*DAPK2*	Death Associated Protein Kinase 2
*DICER1*	Double-Stranded RNA-Specific Endoribonuclease
*FOXF2*	Forkhead Box F2
*GPC3*	Glypican 3
*JAG1*	Jagged Canonical Notch Ligand 1
*JAK1*	Janus Kinase 1
*KLF13*	Kruppel Like Factor 13
*LEFTY2*	Left-Right Determination Factor 2
*LATS2*	Large Tumor Suppressor Kinase 2
*MBNL2*	Muscleblind-Like Protein 2
*MCL1*	Myeloid Cell Leukemia Sequence 1
*MICA*	Major Histocompatibility Complex Class I Chain-Related Protein A
*MMP2*	Matrix Metallopeptidase 2
*NFIB*	Mechanistic Target of Rapamycin Kinase (MTOR), Nuclear Factor I B
*NR4A2*	Nuclear Receptor Subfamily 4 Group A Member 2
*PIK3C2A*	Phosphatidylinositol-4-Phosphate 3-Kinase Catalytic Subunit Type 2 Alpha
*PLCB4*	Phospholipase C Beta 4
*PTK2B*	Protein Tyrosine Kinase 2 Beta
*TCEAL1*	Transcription Elongation Factor A Like 1
*TEAD4*	TEA Domain Transcription Factor 4
*TGFBR2*	Transforming Growth Factor Beta Receptor 2
*TIMP2*	Tissue Inhibitor of Metalloproteinases 2
*TXNIP*	Thioredoxin Interacting Protein
*RASSF1*	Ras Association Domain Family Member 1
*RECK*	Reversion Inducing Cysteine Rich Protein with Kazal Motifs
*SMAD7*	SMAD (small Mothers Against Decapentaplegic) family member 7
*STX12*	Syntaxin 12

## Data Availability

The data presented in this study are available in this article and Appendix A.

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
