# Peer review of "Aberrant Methylation of the Imprinted C19MC and MIR371-3 Clusters in Patients with Non-Small Cell Lung Cancer"

_cancers, 2023, doi:10.3390/cancers15051466_

Round 1

Reviewer 1 Report

The research work entitle “Aberrant Methylation of the Imprinted C19MC and MIR371-3 Clusters in Patients with Non-Small Cell Lung Cancer” has been reported by Laura Boyero et al. Overall the research work is good.  But, there are few corrections to be carried out for improving the manuscript quality.

1.     Expand the NSCLC in keywords.

2.    Numbering section is missing in 2. Like 2.1., 2.2..

3.    Improve the conclusion part.

4.    Provide high quality images

Author Response

Reviewer #1:  The research work entitle “Aberrant Methylation of the Imprinted C19MC and MIR371-3 Clusters in Patients with Non-Small Cell Lung Cancer” has been reported by Laura Boyero et al. Overall the research work is good.  But, there are few corrections to be carried out for improving the manuscript quality.

  1. Expand the NSCLC in keywords.

We agree with the reviewer on this point and have included “non-small cell lung cancer (NSCLC)’ in keywords.

  1. Numbering section is missing in 2. Like 2.1., 2.2.

The authors thank the reviewer for these useful comments. We have numbered the sections in Materials and Methods.

  1. Improve the conclusion part.

Following your indications, we have improved the conclusion.

‘In conclusion, this study demonstrates that the imprinted C19MC and MIR371-3 clusters undergo polycistronic epigenetic regulation that leads to differential tumor expression in patients with NSCLC. These differences, in turn, deregulate the expression of important and common target genes, many of them with a clear oncogenic and regulatory role in this disease, highlighting five genes (FOXF2, KLF13, MICA, TCEAL1 and TGFBR2) that also have a potential prognostic value in NSCLC patients. All these characteristics make the different components of both clusters an interesting target in oncology that needs to be further investigated; it would even be interesting to evaluate the use of methylation agents as an alternative approach to lung cancer therapy. “

  1. Provide high quality images

We sincerely apologize for this issue, we have improved the quality of the Figures 1A, 2, 5, and 7.

Reviewer 2 Report

1. The authors are requested to provide molecular mechanism(s) which is/are associated with aberrant methylation of the imprinted C19MC and MIR371-3 clusters in patients with Non-Small Cell Lung Cancer.

2. The authors are also requested to provide some in vivo experimental data in support of their observations.

Author Response

Reviewer #2: 

  1. The authors are requested to provide molecular mechanism(s) which is/are associated with aberrant methylation of the imprinted C19MC and MIR371-3 clusters in patients with Non-Small Cell Lung Cancer.
  2. The authors are also requested to provide some in vivo experimental data in support of their observations.

We find these comments particularly interesting, although they are not an endpoint of the job. Following the reviewer´s advice, we have included in the discussion section the limitations of the study, which include these two points:

Our study has the limitation that the mechanism underlying the alterated methylation status of the C19MC and MIR371-3 imprinted clusters was not evaluated. Cancer-related gene hypomethylation is common in solid tumors, which could contribute to increased expression of oncogenes. In this case, miRNAs may act as oncomiRs by inhibiting the expression of tumor suppressor genes. DNA methylation is a dynamic process regulated by the action of DNA demethylases and DNA methyltransferases (DNMT)[80]. Alterations in the expression or activity of these enzymes can lead to changes in DNA methylation patterns that can trigger alterations in key genes involved in cancer development. For example, Zhang et al. have reported that DNMT1, DNMT3A, and DNMT3B show frequency alterations in approximately 3% to 5% of lung cancer patients from the cBioPortal datasets[81]. However, further research is needed to validate the mechanisms in detail, e.g. in organoid in vitro and/or in preclinical in vivo models, to fully understand the mechanisms upstream involved in the aberrant methylation of both clusters and to assess the potential therapeutic applications of targeting them. In addition, we identify relevant miRNA-mRNA interactions in NSCLC patients. In other words, the regulation of the expression of these genes can be partly explained by the activity of these miRNAs; nevertheless, it should be noted that additional mechanisms may also be involved in the regulation of these genes, such as deletion, amplification, mutation, fusion and multiple alterations, and even other miRNAs not included in these clusters, as their regulation can be mediated by the action of many miRNAs. Despite these limitations, this study provides important insights into the consequences of these epigenetic alterations in NSCLC patients and highlights potential targets for future research and therapy. On the other hand, another limitation would be the sample size of patients with COPD included in the study. We have analyzed the DNA methylation pattern of the C19MC and MIR371-3 clusters in a group of COPD patients because they have a three- to six-fold increased risk of developing lung cancer, even if they quit smoking[2]. In our study, we found no statistical differences between the control group without lung cancer and COPD. However, sample size is an important consideration because it may affect the accuracy and reliability of the results. A larger size should be considered to conclude the effect of methylation status of both clusters in this disease.

  1. Feng, S.; Jacobsen, S.E.; Reik, W. Epigenetic Reprogramming in Plant and Animal Development. Science 2010, 330, 622–627, doi:10.1126/SCIENCE.1190614.
  2. Zhang, J.; Yang, C.; Wu, C.; Cui, W.; Wang, L. DNA Methyltransferases in Cancer: Biology, Paradox, Aberrations, and Targeted Therapy. Cancers (Basel). 2020, 12, 1–22, doi:10.3390/CANCERS12082123.

Reviewer 3 Report

In the presented manuscript entitled "Aberrant Methylation of the Imprinted C19MC and MIR371-3 2 Clusters in Patients with Non-Small Cell Lung Cancer" authors investigated the methylation profile of 2 miRNAs clusters located in 19q13.42 in lung cancer patients and controls (with and without COPD) using the Illumina Infinium Human Methylation 450 BeadChip. Then the Authors performed the bioinformatic analyses on the publicly available data to evaluate the miRNA-target interactions (miRTar getLink), selected genes being targeted simultaneously by analyzed miRNAs, followed by expression analysis of the target genes in the lung (GEPIA) and gene ontology molecular functions analysis (Panther). Next, the Authors correlated the validated target mRNA-miRNA expression in primary lung tumors (expression data from TCGA, cancerMIRnome tool). Finally, the authors searched for genes associated with cancer progression and shorter OS.

This is a fascinating and valuable study. However, it has many flows, which I hope will be elucidated in the next version of the manuscript.

Major comments:

The major criticism is related to the lack of analysis of selected miRNA levels or the genes targeted by those miRNAs. The bioinformatics analysis demonstrated that 5 of the miRNA target genes: FOXF2, KLF13, 41 MICA, TCEAL1, and TGFBR2 lower expression, were significantly associated with poor overall survival. However, this study did not evaluate those genes' expression level vs. methylation pattern. Thou the conclusion made by the authors, "this study demonstrates that the imprinted C19MC and MIR371-3 miRNA clusters undergo polycistronic epigenetic regulation leading to deregulation of important and common target genes with potential prognostic value in lung cancer," should be supported by some additional surveys or experiments. The expression of the selected genes may be regulated by many miRNAs, not only those located in the studied clusters. Moreover, as the Authors have demonstrated that the miRNAs from the clusters target many genes,  the increased/decreased miRNAs expression (due to the methylation changes) may not alter the expression of the given genes.

The introduction
It would be good to explain why the authors have focused on these clusters. The introduction could be more detailed. While writing on the oncomiRs in NSCLC, without any examples, a table with a summary of miRS previously correlated with cancer progression or outcome would be good [Lines 62-68].

The sentence in lines 71-74 needs to be clarified.

Patients cohort the information on the control groups is missing.
In Supplementary table S1 there is only a number of COPD cases in the lung cancer or control groups, and general information on the COPD/control group is missing. The following information on COPD and control cohort needs to be included: age range (may be more informative than the IQRs) and some clinical information on the patients (for example GOLD classification in case of COPD subjects, causes for hospitalization/diagnostic in the case of the rest of the group).

It might be easier to follow the patient data if Tables S1 and S2 were merged.

Due to information on the COPD cases in both groups (lung cancer and controls), the following question arises - Does the COPD patients' methylation pattern (presented in section 3.1 and Figure 1C) represents the control patients with COPD (n=4) or with lung cancer and COPD (n=20)?

The methods need to be sufficiently described.

In DNA sample section:

1) during what procedures the lung cancer tissue was obtained? And what kind of tissue was analyzed in the case of COPD and control cohorts?

2) How much tissue was used to extract DNA, and how many ng of DNA were submitted to the bisulfite reaction?

3) Was the DNA quality evaluated after bisulfite treatment in the methylation treatment paragraph?

4) Please give more details concerning the bisulfite-modified library preparation and data analysis f.ex. Include data like the average number of reads per sample.

5) In paragraphs related to integrated analysis and correlation: the information on the source of tissues in the miRNA-mRNA analysis and GEPIA should be more detailed. While reading the paragraph one may think that expression analysis was performed in the studied cohort.

6) Were any analyses performed in the laboratory to confirm the expression level of the miRs in the analyzed clusters?

Results:

1) "Patients with lung cancer showed deregulation at 50 miRNAs included in the C19MC and MIR371-3 clusters" -  is this finding based on the from methylation analysis of the studied cohort or from TCGA dataset?

2)  What the "deregulation" means?

3) "Among all miRNAs, miR-520b, miR-520c, miR-520f, miR-526a1, miR-1283-1, and miR-1283-2 showed greater changes in the DNA hypomethylation pattern (Figure 1B, C, and Table 1)." – does the hypomethylation pattern was observed for the cancer samples or controls, or entire cohort?

4)      Which group is presented on the Circos plot (cancer samples, controls, or entire cohort) ?

5)      In Table 1 the information on the mean methylation levels (or %) in analyzed groups could be added.

6)   In the text there is the following information: "We identified a CpG island in the 5' region of the C19MC miRNA cluster (~54,150,000 bp) and another in the MIR371-3 miRNA cluster (~54,270,000 bp) (green lines) (Figure 2C). However, the CpG sites were distributed throughout both clusters (white lines) (Figure 2D). "According to the presented view in Figure 1 ( data from ref. hum genome hg19), the CpG islands are outside of the analyzed clusters, and within the MIR371-3 cluster, there are no CpG sites. Could You explain this discrepancy?

7)      In section 3.3 the methylation profile of the analyzed clusters was compared between cancer and non-tumoral tissue. Then in the Figure 1 pop-up the methylation level compared to the control group. "The DNA-methylation levels of the C19MC and MIR371-3 clusters were consistently hypomethylated in both histological subtypes compared with non-tumoral tissue (Table S3)). Furthermore, these differences were statistically significant for both clusters (C19MC, p < 0.001; MIR371-2, p = 0.030)"

8)      The results comparing the methylation profile in cancer and non-cancerous tissues should be presented separately, as those results seem interesting.

9)      The results presented in Figure 3 - the Y axis is the log2 ratio of methylation in lung tissue? Please add information in the plot and the legend.

10)   Table 5 – the plots are hard to read.

11)    Could You rewrite the following sentences:

# "We constructed a core miRNA–target network to show the experimentally validated miRNA–target interactions for the miRNAs included in the C19MC and MIR371-2 clusters in order to study their functional relevance"

# "49.6% of the 115 target genes showed a median value higher in normal versus at least one of the histologic subtypes of NSCLC, there being significant differences (p < 0.01) in 31 genes: "to make easier to understand.

12)   Were the genes analyzed in section 3.5 taken from the miRNA-target interaction analysis (section 3.4)?

13)   Could You move the list of genes being commonly targeted by different miRNAs to Figure 4, to increase the visibility of this result? A list of gene names in the text makes it hard to read, and usually, such lists are presented in tables, embedded tables in figure panels or supplementary materials.
The same is with the list of genes in Section 3.5.

Author Response

Reviewer #3: 

In the presented manuscript entitled "Aberrant Methylation of the Imprinted C19MC and MIR371-3 2 Clusters in Patients with Non-Small Cell Lung Cancer" authors investigated the methylation profile of 2 miRNAs clusters located in 19q13.42 in lung cancer patients and controls (with and without COPD) using the Illumina Infinium Human Methylation 450 BeadChip. Then the Authors performed the bioinformatic analyses on the publicly available data to evaluate the miRNA-target interactions (miRTar getLink), selected genes being targeted simultaneously by analyzed miRNAs, followed by expression analysis of the target genes in the lung (GEPIA) and gene ontology molecular functions analysis (Panther). Next, the Authors correlated the validated target mRNA-miRNA expression in primary lung tumors (expression data from TCGA, cancerMIRnome tool). Finally, the authors searched for genes associated with cancer progression and shorter OS.

This is a fascinating and valuable study. However, it has many flows, which I hope will be elucidated in the next version of the manuscript.

Major comments:

The major criticism is related to the lack of analysis of selected miRNA levels or the genes targeted by those miRNAs. The bioinformatics analysis demonstrated that 5 of the miRNA target genes: FOXF2, KLF13, 41 MICA, TCEAL1, and TGFBR2 lower expression, were significantly associated with poor overall survival. However, this study did not evaluate those genes' expression level vs. methylation pattern. Though the conclusion made by the authors, "this study demonstrates that the imprinted C19MC and MIR371-3 miRNA clusters undergo polycistronic epigenetic regulation leading to deregulation of important and common target genes with potential prognostic value in lung cancer," should be supported by some additional surveys or experiments. The expression of the selected genes may be regulated by many miRNAs, not only those located in the studied clusters. Moreover, as the Authors have demonstrated that the miRNAs from the clusters target many genes, the increased/decreased miRNAs expression (due to the methylation changes) may not alter the expression of the given genes.

We value the reviewer´s suggestion on this point. We agree that analysis of selected miRNAs and target genes would be good for the study. Unfortunately, we were unable to perform miRNA and gene expression analyses in our cohort because we did not have enough tissue for RNA isolation. However, in our study, we have identified miRNA-target interactions that have been strongly validated previously (not predicted) by other authors. In addition, we have observed the target mRNA-miRNA correlation in lung cancer. Despite all this, we consider that it would be relevant to include a section with the limitations of the study including these points. 

In addition, we identify relevant miRNA-mRNA interactions in NSCLC patients. In other words, the regulation of the expression of these genes can be partly explained by the activity of these miRNAs; nevertheless, it should be noted that additional mechanisms may also be involved in the regulation of these genes, such as deletion, amplification, mutation, fusion and multiple alterations, and even other miRNAs not included in these clusters, as their regulation can be mediated by the action of many miRNAs. Despite these limitations, this study provides important insights into the consequences of these epigenetic alterations in NSCLC patients and highlights potential targets for future research and therapy”.

The introduction

It would be good to explain why the authors have focused on these clusters. 
The introduction could be more detailed. While writing on the oncomiRs in NSCLC, without any examples, a table with a summary of miRS previously correlated with cancer progression or outcome would be good [Lines 62-68].

We appreciate the reviewer's observation, however, we believe that incorporating a table of oncomiRNAs in the introduction section is not pertinent, as the focus of this study is not oncomiRNAs but imprinted clusters of miRNAs.

Nevertheless, we point out 2 important reviews of miRNAs in lung cancer in order to broaden the readers' knowledge:

For a more extensive review of oncomiRs in lung cancer see[9,10]”.

  1. Ghafouri-Fard, S.; Shoorei, H.; Branicki, W.; Taheri, M. Non-Coding RNA Profile in Lung Cancer. Exp. Mol. Pathol. 2020, 114, 104411, doi:10.1016/J.YEXMP.2020.104411.
  2. Zarredar, H.; Ansarin, K.; Baradaran, B.; Shekari, N.; Eyvazi, S.; Safari, F.; Farajnia, S. Critical MicroRNAs in Lung Cancer: Recent Advances and Potential Applications. Anticancer. Agents Med. Chem. 2018, 18, 1991–2005, doi:10.2174/1871520618666180808125459.

The sentence in lines 71-74 needs to be clarified.

The text has been amended as follows:

In addition, alterations in other clusters located in imprinted regions are also attracting interest due to their downstream targets and their involvement in oncogenic and drug resistance mechanisms, such as the chromosome 19 microRNA (C19MC) and MIR371-3 clusters”.

Patient’s cohort – the information on the control groups is missing.

In Supplementary table S1 there is only a number of COPD cases in the lung cancer or control groups, and general information on the COPD/control group is missing. The following information on COPD and control cohort needs to be included: age range (may be more informative than the IQRs) and some clinical information on the patients (for example GOLD classification in case of COPD subjects, causes for hospitalization/diagnostic in the case of the rest of the group).

It might be easier to follow the patient data if Tables S1 and S2 were merged.

Following the reviewer's recommendations, we have merged Table S1 and S2. Regarding the description of continuous variables, given that the cohorts in our study did not fit a normal distribution, it was most convenient to describe them as Median and Interquartile Range. The Median and the Interquartile Range are measures closer to the description itself and do not have as many inferential connotations as the Mean and the Standard Deviation would have on this occasion. On the other hand, COPD classification is not included in the tables because many patients did not have this information in their medical records.

Due to information on the COPD cases in both groups (lung cancer and controls), the following question arises - Does the COPD patients' methylation pattern (presented in section 3.1 and Figure 1C) represents the control patients with COPD (n=4) or with lung cancer and COPD (n=20)?

We have detailed the comparison carried out in the section 3.1., and Figure 1 C to clarify the text according to the reviewer's commentary:

In the section 3.1:

“The methylation levels in lung cancer versus paired non-tumoral tissue are represented in Figure 1A and Supplementary Table S3. Patients with lung cancer at our hospital showed DNA hypomethylation at 50 miRNAs included in the C19MC and MIR371-3 clusters after standardisation with non-tumoral control samples. Statistically significant differences (adjusted p-value < 0.05) were detected in the large C19MC cluster (46 miRNAs) and the closely distal MIR371–3 cluster (miR-371a, miR-371b, miR-372 and miR-373). Among all miRNAs, miR-520b, miR-520c, miR-520f, miR-526a1, miR-1283-1, and miR-1283-2 showed greater changes in the DNA hypomethylation pattern in NSCLC patients (Figure 1B, C, and Table 1).”

In the legend of figure 1:

Figure 1. Methylation profiles of the C19MC and MIR371-3 miRNA clusters in lung cancer. A) Circos plot showing the methylation levels on chromosome 19. From inside to outside: methylation levels, ideogram and gene labels. Hypermethylation (red dots and green background) and hypomethylation events (green dots and red background) in patients with lung cancer versus paired non-tumoral samples. NA: Not available. B) Observed methylation changes (log2 ratio) in the C19MC cluster. Relative levels of methylation in patients with lung cancer relative to the control group are represented in blue bars, whereas methylation levels of COPD patients without lung cancer compared to the non-tumoral control group are represented by red bars. A grey background represents statistically significant differences (adjusted p-value < 0.05) of methylation levels relative to the control group. C) Observed methylation changes (log2 ratio) in the MIR371-3 cluster. Relative levels of methylation in patients with lung cancer relative to the control group are represented in blue bars, whereas methylation levels of COPD patients without lung cancer compared to the non-tumoral control group are represented by red bars. A grey background represents statistically significant differences (adjusted p-value < 0.05) of methylation levels in comparison to the non-tumoral control group.

The methods need to be sufficiently described.

In DNA sample section:

During what procedures the lung cancer tissue was obtained? And what kind of tissue was analyzed in the case of COPD and control cohorts

 Following your indications we have improved the information included in the section 2.1:

“… A first cohort consisted of 47 NSCLC patients who had undergone surgical resection at an early clinical stage.”

This control cohort consisted of COPD patients and non-COPD subjects (Supplementary Table S1) who had undergone bullectomy or bronchoscopy biopsy with a negative diagnosis of lung cancer”

 How much tissue was used to extract DNA, and how many ng of DNA were submitted to the bisulfite reaction?

Following the reviewer´s suggestion, we have modified the section 2.2 (DNA sample):

Genomic DNA was extracted from 15mg adjacent normal and tumor tissue samples using the QIAamp DNA mini kit (QIAGEN, CA, USA).”

In the section 2.3 (Illumina 450K methylations assay) is already included how many ng of DNA were treated with sodium bisulfate:

500 ng of DNA were treated with sodium bisulfate using the EZ DNA Methylation™ Kit and cleaned with the ZR-96 DNA Clean-up Kit™ (Zymo Research, Irvine, CA).”

3) Was the DNA quality evaluated after bisulfite treatment in the methylation treatment paragraph?

The HumanMethylation450 BeadChip includes 600 negative controls, which are particularly important in methylation analysis assays since sequence complexity is decreased after bisulfite conversion. Moreover, the GenomeStudio® Methylation Module Software has an integrated Controls Dashboard where the performance of all controls can be easily monitored.

We have included in the section 2.3 (Illumina 450K methylation assay) that:

“Infinium HD-based assays included sample-dependent and sample-independent controls for the highest quality data.”

4) Please give more details concerning the bisulfite-modified library preparation and data analysis f.ex. Include data like the average number of reads per sample.

We would like to clarify that the assay has been done by methylation array and not by sequencing. Thus, Illumina Infinium HumanMethylation450 BeadChip (450K) (https://www.illumina.com/content/dam/illumina-marketing/documents/products/datasheets/datasheet_humanmethylation450.pdf) is based in single base extension of a specific probe corresponding to a specific CpG coupled to different fluorescent dyes in an array. This array is quantified by fluorescence intensity and therefore, it is not possible include the average number of reads per samples as obtained with other technologies with genomic libraries.

 5) In paragraphs related to integrated analysis and correlation: the information on the source of tissues in the miRNA-mRNA analysis and GEPIA should be more detailed. While reading the paragraph one may think that expression analysis was performed in the studied cohort.

We appreciate the reviewer's suggestion and we have modified the text to include clarifications on the data sources used by each of the bioinformatics tools.

“Strong validated miRNA-mRNA interactions were identified for the C19MC and MIR371-3 miRNA clusters with the miRTargetLink 2.0 Human tool. miRTargetLink collects information from various databases, including the sources miRBase (v.22.1) and miRTarBase (v.8)[36,37]. Gene expression analysis on tumor and normal lung tissue were analyzed from the Cancer Genome Atlas datasets (TCGA) using GEPIA 2.0 (http://gepia2.cancer-pku.cn/#index). GEPIA is an interactive web server that compiles data from TCGA and GTEx projects, using a standard processing pipeline. This tool allows for a customizable analysis of the collected data[38]. The molecular function and proposed biological process of the experimentally verified mRNAs were determined using the PANTHER program (http://pantherdb.org). The PANTHER classification system contains a comprehensive, annotated "library" of phylogenetic trees of gene families designed to classify proteins (and their genes) to facilitate high-throughput analysis[39]. Besides, Kyoto Encyclopedia of Genes and Genomes (KEGG) pathway database was used to know the molecular interaction network (https://www.genome.jp/kegg/pathway.html)[40]”.

6) Were any analyses performed in the laboratory to confirm the expression level of the miRs in the analyzed clusters?

As previously commented to the reviewer, we could not perform expression analysis of miRNAs because we did not have sample for RNA isolation.

Results:

1) "Patients with lung cancer showed deregulation at 50 miRNAs included in the C19MC and MIR371-3 clusters" -  is this finding based on the from methylation analysis of the studied cohort or from TCGA dataset?

Given the ambiguity of the phrase, it has been modified. The final version is the following:

“To evaluate the potential role of the C19MC and MIR371-3 miRNA clusters in lung cancer, we analyzed the methylation status of both of them in human lung tissues from a NSCLC patient cohort (N = 47) and a control cohort (N = 23) of the Virgen del Rocio University Hospital (Seville, Spain)”.

2)  What the "deregulation" means?

For greater clarification we have replaced “deregulation” with “DNA hypomethylation”.

3) "Among all miRNAs, miR-520b, miR-520c, miR-520f, miR-526a1, miR-1283-1, and miR-1283-2 showed greater changes in the DNA hypomethylation pattern (Figure 1B, C, and Table 1)." – does the hypomethylation pattern was observed for the cancer samples or controls, or entire cohort?

As indicated in the figure caption and table title, blue bars represent the hypomethylation levels of cancer samples relatives to control samples. Red bars represent the methylation levels of COPD samples without lung cancer relatives to non-tumoral control samples.

For clarification, the text has been modified to include that methylation levels were normalized to control samples for representation.

Patients with lung cancer at our hospital showed deregulation DNA hypomethylation at 50 miRNAs included in the C19MC and MIR371-3 clusters after standardisation with non-tumoral control samples”.

4)      Which group is presented on the Circos plot (cancer samples, controls, or entire cohort) ?

For greater clarity, the text has been amended:

In section 3.1.:

The methylation levels in lung cancer versus paired non-tumoral tissues are represented in Figure 1A and Supplementary Table S2”.

In Fig 1A caption:

Hypermethylation (red dots and green background) and hypomethylation events (green dots and red background) in patients with lung cancer versus paired non-tumoral samples”.

5)      In Table 1 the information on the mean methylation levels (or %) in analyzed groups could be added.

Following the reviewer's recommendations, the relative changes in methylation (log2) have been included in Table 1.

6)   In the text there is the following information: "We identified a CpG island in the 5' region of the C19MC miRNA cluster (~54,150,000 bp) and another in the MIR371-3 miRNA cluster (~54,270,000 bp) (green lines) (Figure 2C). However, the CpG sites were distributed throughout both clusters (white lines) (Figure 2D). "According to the presented view in Figure 1 ( data from ref. hum genome hg19), the CpG islands are outside of the analyzed clusters, and within the MIR371-3 cluster, there are no CpG sites. Could You explain this discrepancy?

We find this comment particularly interesting. CpG islands are typically common near transcription start sites and may be associated with promoter regions. Following the reviewer´s advice, we have improved the quality of Figure 2 so that the two CpG islands at the beginning of both clusters can be seen without problem (Figure 2C). In addition, we have included the following text in the results section 2.2:  

“We identified a CpG island in the 5' region at the beginning of the C19MC miRNA cluster (~54,150,000 bp; CpG count:86; and citosine base count plus guanine base count:762) and another in the MIR371-3 miRNA cluster (~54,270,000 bp; CpG count:23; and citosine count plus guanine count:157) (green lines) (Figure 2C).” 

7)      In section 3.3 the methylation profile of the analyzed clusters was compared between cancer and non-tumoral tissue. Then in the Figure 1 pop-up the methylation level compared to the control group. "The DNA-methylation levels of the C19MC and MIR371-3 clusters were consistently hypomethylated in both histological subtypes compared with non-tumoral tissue (Table S3)). Furthermore, these differences were statistically significant for both clusters (C19MC, p < 0.001; MIR371-2, p = 0.030)"

We would like to clarify that figure 1 shows the methylation levels of lung cancer versus paired non-tumoral samples and COPD without lung cancer versus non-tumoral samples. Once it was proven that NSCLC samples are hypomethylated, in section 3.3 we studied whether there were differences between histological subtypes.

To avoid ambiguity, we modify the text as follows:

“…we analyzed the methylation pattern in SCC versus adenocarcinoma, pre-normalising each patient with the methylation status of the matched non-tumoral tissue”.

 8)      The results comparing the methylation profile in cancer and non-cancerous tissues should be presented separately, as those results seem interesting.

We appreciate the reviewer's comment, however, the requested results are already represented in Fig 1 and described in section 3.1.

“…The methylation levels in lung cancer versus paired non-tumoral tissues are represented in Figure 1A and Supplementary Table S2. Patients with lung cancer at our hospital showed DNA hypomethylation at 50 miRNAs included in the C19MC and MIR371-3 clusters after standardisation with non-tumoral control samples…”.

9)      The results presented in Figure 3 - the Y axis is the log2 ratio of methylation in lung tissue? Please add information in the plot and the legend.

Following the reviewer's recommendations, we have included in the Figure 3 caption:

Relative methylation changes in β values (log2) are represented on the y-axis”.

10)   Table 5 – the plots are hard to read.

We sincerely apologize for this issue, we have improved the quality of the Figure 5.

11)    Could You rewrite the following sentences:

# "We constructed a core miRNA–target network to show the experimentally validated miRNA–target interactions for the miRNAs included in the C19MC and MIR371-2 clusters in order to study their functional relevance"

We have modified the text as follows:

We have graphically represented a network with experimentally validated miRNA-mRNA interactions for each component of the C19MC and MIR371-3 clusters in order to study their functional relevance”.

# "49.6% of the 115 target genes showed a median value higher in normal versus at least one of the histologic subtypes of NSCLC, there being significant differences (p < 0.01) in 31 genes: "to make easier to understand.

We have modified the text as follows:

Of the 115 target genes validated by miRTargetLink, 31 were significantly underexpressed in at least one of the histological subtypes of NSCLC compared to non-tumor tissue (Table 3 and Figure 5)”.

12)   Were the genes analyzed in section 3.5 taken from the miRNA-target interaction analysis (section 3.4)?

The answer is yes, the genes analysed in section 3.5 taken from the miRNA-target interaction analysis (section 3.4). For clarification, we have modified the text as follows:“Of the 115 target genes validated by miRTargetLink, 31 were significantly underexpressed in at least one of the histological subtypes of NSCLC compared to non-tumor tissue (Table 3 and Figure 5)”.

13)   Could You move the list of genes being commonly targeted by different miRNAs to Figure 4, to increase the visibility of this result? A list of gene names in the text makes it hard to read, and usually, such lists are presented in tables, embedded tables in figure panels or supplementary materials.
The same is with the list of genes in Section 3.5.

Following the reviewer's recommendations, Tables 2 and 3 have been prepared.

Reviewer 4 Report

Thank you for the chance you gave me to read this interesting study entitled “Aberrant Methylation of the Imprinted C19MC and MIR371-3 2 Clusters in Patients with Non-Small Cell Lung Cancer” by Boyero et al. In this study, the authors evaluated methylation profile of the imprinted C19MC and MIR371-3 clusters in patients with non-small cell lung cancer (NSCLC) as well as their potential target genes and their prognostic role. DNA methylation status was analyzed in 47 NSCLC patients and 23 controls consisted of COPD patients and non-COPD subjects. Interestingly, hypomethylation of miRNAs located on chromosome 19q13.42 was found to be specific for tumor tissue (more accentuated in SCC than in lung adenocarcinoma) as well as that a lower expression of 5 of the target genes (FOXF2, KLF13, 41 MICA, TCEAL1, and TGFBR2) had a significantly negative prognostic value. The study is well-designed and organized as well as well presented. I think that this study in the current form does satisfy the appropriate criteria for publication in your journal, however, some minor points should be treated before being suitable for publication.

Some of them are:

According to the plagiarism detection service “Turnitin”, this study has a high score (36%). The authors could improve this issue.

Abbreviations should be expanded at first mention e.g. NF-κB.

Control group should be described better in the abstract.

Although the manuscript is generally well written, however, it needs to be literally edited for some minor mistakes and strange structures.

Figure 1: Green and red dots are indiscernible.

In figure 3, differences between boxplots (ADC vs SCC) are not correct, since in the text the authors mention that DNA-methylation levels in lung cancer patients were compared with non-tumoral tissue. Please, double check.

Limitation paragraph should be added including some methodological issues (e.g differences in COPD patients enrollment in two cohorts, should COPD patients be included in the control group? etc).

Author Response

Reviewer #4:

Thank you for the chance you gave me to read this interesting study entitled “Aberrant Methylation of the Imprinted C19MC and MIR371-3 2 Clusters in Patients with Non-Small Cell Lung Cancer” by Boyero et al. In this study, the authors evaluated methylation profile of the imprinted C19MC and MIR371-3 clusters in patients with non-small cell lung cancer (NSCLC) as well as their potential target genes and their prognostic role. DNA methylation status was analyzed in 47 NSCLC patients and 23 controls consisted of COPD patients and non-COPD subjects. Interestingly, hypomethylation of miRNAs located on chromosome 19q13.42 was found to be specific for tumor tissue (more accentuated in SCC than in lung adenocarcinoma) as well as that a lower expression of 5 of the target genes (FOXF2, KLF13, 41 MICA, TCEAL1, and TGFBR2) had a significantly negative prognostic value. The study is well-designed and organized as well as well presented. I think that this study in the current form does satisfy the appropriate criteria for publication in your journal, however, some minor points should be treated before being suitable for publication.

Some of them are:

  1. According to the plagiarism detection service “Turnitin”, this study has a high score (36%). The authors could improve this issue.

We are very surprised by this comment from the reviewer. We have accessed the Turnitin software to check the plagiarism percentage and found a 100% match, so we suspect that the reviewer uploaded our manuscript to Turnitin and has not yet removed it. Thus, when we upload our file, the software detects that it is identical to the one already uploaded by the reviewer.

This prevents us from doing our own checking in Turnitin. Therefore, we have used the free website https://www.check-plagiarism.com/ to check it and we have verified that we have a 2% plagiarism rate.

We believe that the 36% plagiarism detected by the reviewer could be due to the references section.

  1. Abbreviations should be expanded at first mention e.g. NF-κB.

All the text has been reviewed for this matter and the pertinent modifications have been made.

  1. Control group should be described better in the abstract.

We have modified the text according to the reviewer’s suggestion. The final version is:

DNA methylation status was analyzed in a NSCLC patient cohort (N = 47) and compared with a control cohort cohort including of COPD patients and non-COPD subjects (N = 23) using the Illumina Infinium Human Methylation 450 BeadChip”.

  1. Although the manuscript is generally well written, however, it needs to be literally edited for some minor mistakes and strange structures.

Thank you for your comment. All text has been revised for improvement.

  1. Figure 1: Green and red dots are indiscernible.

We sincerely apologize for this issue, we have improved the quality of the Figure 1.

  1. In figure 3, differences between boxplots (ADC vs SCC) are not correct, since in the text the authors mention that DNA-methylation levels in lung cancer patients were compared with non-tumoral tissue. Please, double check.

ADC and SCC data are plotted relative to the control group, which means that the data have been pre-standardised. To clarify this, the text in section 3.3. has been modified:

Due to lung cancer-specific hypomethylation, in order to evaluate the potential role of the C19MC and MIR371-3 miRNA clusters as biomarkers in different histological subtypes of lung cancer, we analyzed the methylation pattern in SCC versus adenocarcinoma, pre-normalising each patient with the methylation status of the matched non-tumoral tissue.”.

  1. Limitation paragraph should be added including some methodological issues (e.g differences in COPD patients enrollment in two cohorts, should COPD patients be included in the control group? etc).

We appreciate the reviewer's comment. The following paragraph has been added to the study limitations section:

On the other hand, another limitation would be the sample size of patients with COPD included in the study. We have analyzed the DNA methylation pattern of the C19MC and MIR371-3 clusters in a group of COPD patients because they have a three- to six-fold increased risk of developing lung cancer, even if they quit smoking[2]. In our study, we found no statistical differences between the control group without lung cancer and COPD. However, sample size is an important consideration because it may affect the accuracy and reliability of the results. A larger size should be considered to conclude the effect of methylation status of both clusters in this disease”.

Round 2

Reviewer 3 Report

The work has been improved to a significant extent. In the current version, it is much more evident.